# SpecExtend: A Drop-in Enhancement for Speculative Decoding of Long Sequences

## Abstract

Speculative decoding is a widely used technique for accelerating inference in large language models (LLMs), but its performance degrades as input length grows, with significant drops even at moderate lengths. Yet, this early degradation has remained largely underexplored. We introduce SpecExtend, a drop-in enhancement that improves speculative decoding on long sequences without additional training. SpecExtend integrates efficient attention mechanisms such as FlashAttention and Hybrid Tree Attention to accelerate prefill and verification steps. To improve both draft accuracy and speed on long inputs without retraining, we propose Cross-model Retrieval, a novel KV cache eviction strategy that leverages the target model's attention scores to dynamically select relevant context for the smaller draft model. Extensive evaluations show that SpecExtend accelerates speculative decoding by up to 2.84× on 16K-token long summarization and up to 3.86× on long reasoning, while preserving the short-input performance of state-of-the-art frameworks.

## 1 Introduction

Large Language Models (LLMs) have achieved remarkable success across a wide range of natural language processing (NLP) tasks. However, their practical deployment is often hindered by high inference latency, which is primarily caused by the autoregressive nature of decoding. To address this issue, various optimization techniques have been proposed, with speculative decoding emerging as an effective, lossless solution. Speculative decoding consists of two phases: First, a smaller draft model is used to efficiently generate multiple candidate tokens. Then, the original target model verifies these tokens in parallel. This allows generating multiple tokens within a single target model decoding step, accelerating inference without altering the output distribution.

Despite these advantages, the performance of speculative decoding frameworks drops significantly as input length increases. When the input becomes extremely long, the memory bottleneck shifts from model weights to the KV cache. Prior work (Sun et al., 2024; Sadhukhan et al., 2024) has attempted to address this by using sparse KV caches of the target model for drafting. As shown in Figure 2, however, performance degradation arises much earlier than this bottleneck shift, and existing methods yield little speedup due to drafting with the slow base model that has large weights. Yet, this degradation in the moderate-length regime is largely underexplored. We identify two main causes: (1) increased latency in the forward passes of both target and draft models due to the quadratic complexity of standard attention, and (2) reduced draft accuracy, as the draft model is typically smaller and trained only on short sequences. To address this, a drop-in solution is desirable, since retraining draft models on long contexts is costly, while tasks like long-form generation begin with short inputs and gradually expand, requiring the solution to preserve short-input performance and the original benefits of existing state-of-the-art frameworks.

The theoretical speedup of speculative decoding (Equation 1) shows that in the moderate-length regime, it is critical to maintain high draft accuracy, as it reduces the total number of verification steps required. A simple way to improve draft accuracy without retraining is to shrink the draft model's KV cache with an eviction policy such as StreamingLLM (Xiao et al., 2023), known to improve both generation quality and speed on long inputs. However, with such a static eviction policy, draft accuracy still degrades when tasks require finer-grained use of past context (e.g., Needle Retrieval), due to the loss of important context.

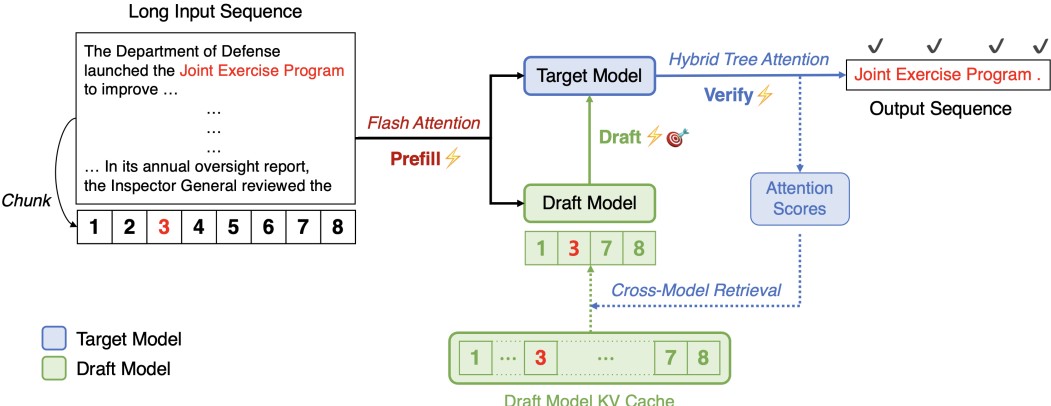

Figure 1: Overview of SpecExtend. FlashAttention accelerates the prefill phases of both target and draft models, and Hybrid Tree Attention accelerates the verification phase. We use the target model's attention scores obtained from verification to select the most relevant input chunks to retain in the draft model's KV cache, enhancing both draft speed and accuracy on long inputs.

To this end, we propose SpecExtend, a drop-in enhancement for speculative decoding on long inputs (Figure 1). We first incorporate efficient attention mechanisms (Section 3.1) such as FlashAttention and Hybrid Tree Attention to accelerate the prefill and verification steps. To improve draft accuracy and speed without retraining, we introduce Cross-model Retrieval (Section 3.2), a novel cache update strategy for speculative decoding. We dynamically update the smaller draft model's KV cache with globally relevant context, guided by the larger target model's attention scores. By enabling fine-grained alignment between draft and target models in long contexts, this improves the average accepted length by up to 2.55× on inputs of up to 16K tokens, outperforming static eviction strategies.

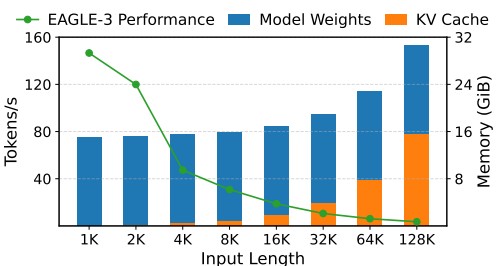

Figure 2: Performance and memory usage of speculative decoding with Llama-3.1-8B-Instruct and EAGLE-3 across varying input lengths. Performance significantly declines well before the shift of memory bottleneck.

We evaluate SpecExtend on practical long-sequence generation tasks where speculative decoding typically struggles, using both off-the-shelf LLMs and EAGLE draft models. On long summarization with inputs of up to 16K tokens (GovReport, PG-19, BookSum), SpecExtend achieves up to 2.22× speedup with Vicuna-7B and 2.84× with Llama-3.1-8B-Instruct. On long reasoning (AIME-24), it yields up to 3.86× speedup with DeepSeek-R1-Distill-Llama-8B. SpecExtend is compatible with various speculative decoding setups and robust across input lengths. Importantly, it is training-free and preserves short-input performance, enabling the use of powerful state-of-the-art frameworks such as EAGLE-3 for long-sequence generation.

Our main contributions are as follows:

- To the best of our knowledge, we are the first to tackle the largely underexplored problem of speculative decoding performance degradation in the moderate-length regime with a training-free solution.

- We propose *Cross-model Retrieval*, a novel KV cache eviction strategy that improves both draft accuracy (by up to 2.3×) and speed on long inputs, without additional training. It consistently outperforms static cache eviction policies, and we provide in-depth analysis of its effectiveness.

- We introduce *SpecExtend*, a drop-in solution that accelerates speculative decoding by up to 2.22× on 16K-token long summarization and up to 3.86× on long reasoning, while preserving the short-input performance of state-of-the-art frameworks.

## 2 RELATED WORK

**Speculative Decoding** Speculative decoding accelerates LLM inference by using a smaller draft model to generate multiple candidate tokens, which the target model then verifies in parallel (Xia et al., 2022; 2024). With proper verification and correction, it guarantees the same output distribution as standard decoding (Leviathan et al., 2023; Chen et al., 2023). SpecInfer (Miao et al., 2024) extends this approach by drafting and verifying multiple sequences simultaneously using tree attention, achieving further speedups. Several works introduce effective draft models built from subsets of the target model (Cai et al., 2024; Li et al., 2024b), while EAGLE-2 (Li et al., 2024c) and OPT-Tree (Wang et al., 2025) achieve further speedup by dynamically adjusting the draft tree structure during decoding. EAGLE-3 (Li et al., 2025) scales up draft model training by leveraging multi-level features from the target model.

**Long Sequence Generation** As input length increases, standard attention suffers from quadratic computational and memory complexity, causing high inference latency (Zhou et al., 2024). FlashAttention (Dao et al., 2022; Dao, 2023) reduces this overhead by using tiling and online softmax, bringing memory complexity down to linear and accelerating inference. FlashDecoding (Dao, 2024) builds on this by further parallelizing workers across the Key-Value dimension, speeding up LLM decoding for long sequences.

Several works apply speculative decoding to long sequence generation. Sadhukhan et al. (2024) identify that the memory bottleneck shifts from model weights to the KV cache for extremely long inputs, and use sparse KV cache of the base model to draft tokens. Sun et al. (2024) mitigate this with hierarchical speculation using both a smaller draft model and sparse KV cache of the base model. However, the performance of speculative decoding frameworks drop well before the KV cache becomes the main bottleneck, and existing solutions yield marginal speedup in this regime of early degradation. Closest to our approach is LongSpec (Yang et al., 2025), which trains draft models specifically designed for long inputs. In contrast, our method provides a drop-in enhancement for existing frameworks, improving long-sequence performance without retraining while preserving their original benefits, such as short-input performance.

## 3 SPECEXTEND

We first give an overview of SpecExtend's components: efficient attention mechanisms that accelerate forward passes (Section 3.1) and Cross-model Retrieval that enhances both draft speed and accuracy without additional training (Section 3.2). We then provide the theoretical speedup analysis (Section 3.2.2) and an in-depth analysis on the effectiveness of Cross-model Retrieval (Section 3.2.3).

### 3.1 EFFICIENT ATTENTION

Standard attention becomes impractical with longer inputs due to its quadratic complexity, making it essential to incorporate efficient attention mechanisms. The initial forward pass of LLM inference, known as the prefill stage, computes full self-attention over the entire input sequence, incurring quadratic memory usage and latency. FlashAttention (Dao et al., 2022; Dao, 2023) mitigates this by avoiding materialization of large intermediate matrices in the GPU high-bandwidth memory. We apply FlashAttention to the prefill stages of both the target and draft models, reducing latency and memory usage during this phase (Figure 1).

Unlike prefill, the decoding stage uses cached KV states and computes attention only with the newly generated tokens as query. FlashDecoding (Dao, 2024) accelerates this step by additionally parallelizing across the KV sequence length. Meanwhile, Hybrid Tree Attention allows FlashDecoding to be compatible with the tree-structured attention required in modern speculative decoding frameworks (Yang et al., 2025). We apply Hybrid Tree Attention to the target model to accelerate the verification step of speculative decoding.

## 3.2 CROSS-MODEL RETRIEVAL

### 3.2.1 METHOD OVERVIEW

As input length increases, draft speed in standard speculative decoding degrades because the draft model's KV cache grows, leading to slower decoding. Meanwhile, draft accuracy also drops due to the draft model's limited capacity as it is much smaller than the base model and typically trained on short contexts. To address this without retraining, we aim to truncate the draft model's KV cache for more efficient attention, while preserving context that is most relevant to the target model at the current decoding timestep. We achieve this via *Cross-model Retrieval* (CMR), which uses the target model's attention scores to select the most relevant input chunks to retain in the smaller draft model's cache. The procedure is detailed in Algorithm 1.

Concretely, we divide the input prefix into fixed-size chunks and rank them by their average attention scores, using the last accepted token as the query. These scores reflect each chunk's relevance at the current timestep. We select the top-k chunks, and the draft model uses this reduced, fine-grained cache to generate candidate tokens, enhancing both draft speed and accuracy on long inputs.

Importantly, the target model's attention scores are obtained directly from the most recent verification step, requiring no additional forward passes. One challenge is that the target model's Hybrid Tree Attention relies on FlashDecoding, which avoids generating the full attention scores matrix for efficiency. To address this, we compute standard attention and extract attention scores of only the final layer, which we find sufficient for our purposes. As shown in Table 8, this adds minimal latency overhead to the target model's forward pass, and the cache update step is also faster than a single draft model forward pass. Moreover, due to the locality of context in long sequences, retrieval cache updates can be applied adaptively or less frequently, further minimizing overhead.

---

**Algorithm 1** Speculative Decoding with Cross-model Retrieval

**Require:** Target LM $M_q$, draft LM $M_p$, input $x_1, \ldots, x_t$, block size $K$, target length $T$, DRAFT, VERIFY, CORRECT, retrieval flag doRetrieval, attention scores $s$, top-$k$ chunks $c_1, \ldots, c_k$

1: $n \leftarrow t$
2: **while** $n < T$ **do**
      ▷ Retrieve and update draft model cache
3:   **if** doRetrieval **then**
4:     $c_1, \ldots, c_k \leftarrow$ SELECTCHUNKS$(s)$
5:     UPDATEDRAFTCACHE$(c_1, \ldots, c_k)$
6:   $p_1, \ldots, p_K \leftarrow$ DRAFT$(x_{\leq n}, M_p)$
7:   Sample $\tilde{x}_i \sim p_i$ for $i = 1, \ldots, K$
      ▷ Obtain target model attention scores for $i = 1, \ldots, K+1$
8:   $(q_i, s)$
        $\leftarrow M_q\big(x \mid x_{\leq n}, \tilde{x}_{<i} \,;\, \text{doRetrieval}\big)$
9:   **if** VERIFY$(\tilde{x}_i, p_i, q_i)$ **then**
10:    $x_{n+1} \leftarrow \tilde{x}_i; n \leftarrow n + 1$
11:  **else**
12:    $x_{n+1} \leftarrow$ CORRECT$(p_i, q_i)$
13:    **break**
14:  **if** all $K$ drafted tokens accepted **then**
15:    Sample $x_{n+1} \sim q_{K+1}; n \leftarrow n + 1$

---

### 3.2.2 THEORETICAL SPEEDUP ANALYSIS

Equation 1 formalizes the speedup of standard speculative decoding (Sadhukhan et al., 2024), where $T_t$ denotes the target model's per-token latency, $T_d$ the draft model's per-token latency, $T_v$ the verification cost, and $\tau$ the average accepted length. Speedup is achieved only when drafting is sufficiently fast, that is, when $T_d$ is small compared to $T_t$. Figure 2 shows that in the moderate-length regime, model weights remain the dominant memory bottleneck even as the KV cache grows. Thus, improving draft speed in this regime requires reducing both model weights and KV cache size, which existing methods fail to achieve. At the same time, it is critical to maintain high draft accuracy, which leads to higher $\tau$.

$$\frac{T_{avg}^{sd}}{T_t} = \frac{1}{\tau(n,d)} \left( \frac{d \cdot T_d}{T_t} + \frac{T_v(n)}{T_t} \right) \tag{1}$$

SpecExtend addresses both requirements in a training-free manner: it substantially reduces $T_d$ by employing a smaller draft model *and* a reduced KV cache, while preserving draft accuracy by retaining the most important information via CMR. Moreover, SpecExtend's efficient attention mechanisms further improve end-to-end speedup on long inputs: FlashAttention reduces prefill time which otherwise dilutes the overall speedup, while Hybrid Tree Attention accelerates verification and reduces $T_v$.

### 3.2.3 IN-DEPTH ANALYSIS OF EFFECTIVENESS

**Needle Retrieval Evaluation**   Cross-model Retrieval reduces the draft model's KV cache by selecting chunks ranked with the attention scores of a much larger base model. This raises a key question: *Even if the retrieved chunks are optimal according to the base model, can the smaller draft model actually leverage them to draft tokens more accurately?* To answer this, we use the Needle Retrieval task and measure how well the draft model uses the retrieved context to identify and generate tokens corresponding to a planted "needle" in long inputs (Li et al., 2024a; Contributors, 2023). We compare its accuracy against three draft model cache strategies: (1) **Full KV Cache** which retains all context; (2) **StreamingLLM** (Xiao et al., 2023) which keeps only the earliest and most recent tokens via a static cache policy; and (3) **TriForce** (Sun et al., 2024) which also retrieves top chunks using the base model's attention scores but performs both drafting and verification with the large base model itself. While accurate, drafting with the base model is slow in the moderate-length regime due to its large weights. Therefore, TriForce serves as a reference for the ideal case on how well retrieved context can be utilized by a much smaller draft model.

| Cache Type | Full KV Cache | StreamingLLM | Cross-model Retrieval (SpecExtend) | Retrieval (TriForce) |
|---|---|---|---|---|
| Draft Model Size | 160M | 160M | 160M | 7B |
| Perplexity ($\downarrow$) | 8.311 | 2.435 | 2.237 | 2.191 |
| Accuracy ($\uparrow$) | 0.081 | 0.166 | 0.823 | 0.976 |

Table 1: Perplexity and draft accuracy of needle tokens in the Needle Retrieval task with Vicuna-7B/160M. TriForce uses Vicuna 7B for both drafting and verification.

As shown in Table 1, while StreamingLLM improves general coherence, it struggles to draft the needle tokens accurately due to loss of global context. In contrast, CMR approaches TriForce's performance despite using a smaller draft model, simultaneously enhancing draft speed *and* accuracy for long inputs. This demonstrates the draft model's potential to utilize fine-grained context retrieved by a much larger model.

**Accuracy and Divergence Analysis**   We further examine token types that benefit from CMR during drafting. We measure the distribution entropy of generated tokens, where higher entropy indicates harder or more informative tokens. Tokens in the top 10% of entropy are classified as *hard*, and we compare their acceptance rates under StreamingLLM and CMR. While the Needle Retrieval evaluation suggests that CMR helps primarily with hard tokens, Figure 3 shows that it improves draft accuracy for both hard and easy tokens. We also measure the natural divergence (Leviathan et al., 2023) between the draft and target models across accepted and resampled token positions. Figure 3 demonstrates that CMR consistently yields lower divergence at all positions.

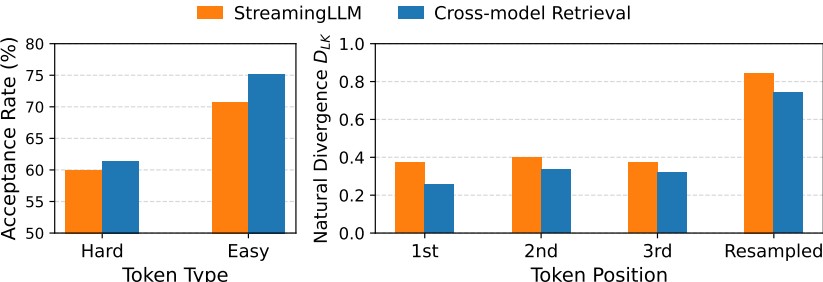

Figure 3: Left figure shows acceptance rates for hard and easy tokens, where CMR enables more accurate drafting in both cases compared to StreamingLLM. Right figure shows the natural divergence between the target and draft models at the first three accepted tokens and the resampled token. CMR consistently yields lower divergence across all positions.

This indicates that by supplying the draft model with target-guided, fine-grained context, CMR shifts its distribution closer to the target not only for hard tokens but also for frequent, easier ones, compared to StreamingLLM. Thus, CMR extends its benefit beyond recovering needles, broadly enhancing draft–target alignment in long contexts and general tasks. We further provide an ablation study of CMR's performance against StreamingLLM on long summarization (Section 4.3.1).

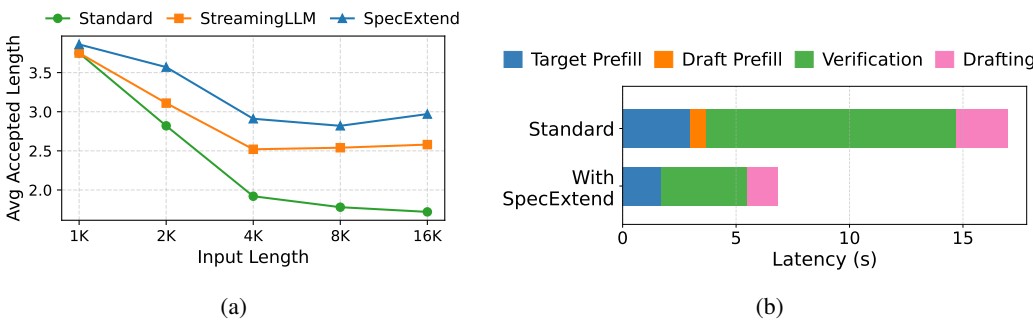

(a)  (b)

Figure 4: (a) Average accepted length of Vicuna-7B/68M across different draft model cache settings. (b) End-to-end latency breakdown of speculative decoding on 16K-token inputs.

## 4 EXPERIMENTS

**Experiment Setup**   We evaluate SpecExtend on two practical long-sequence generation tasks with distinct characteristics, both of which pose challenges for standard speculative decoding: (1) **long summarization**, where the model processes a very long input from the start, and (2) **long reasoning**, where the input is short but the generated output grows very long.

For long summarization, we use Vicuna-7B-16K (Chiang et al., 2023) and LongChat-7B-16K (Li et al., 2023) as base models, with both EAGLE (Li et al., 2024b) and off-the-shelf LLMs, Vicuna-68M/LLaMA-68M (Yang et al., 2024b; Miao et al., 2024) as draft models. We adopt tree-based drafting with dynamic tree expansion (Miao et al., 2024; Wang et al., 2025) and evaluate on Gov-Report (Huang et al., 2021), PG-19 (Rae et al., 2019), and BookSum (Kryściński et al., 2021), generating 256 tokens with temperature 0. For long reasoning, we use DeepSeek-R1-Distill-Llama-8B (DeepSeek-AI, 2025) as the base model and EAGLE-3 (Li et al., 2024b) as the draft model. We evaluate on the AIME-24 benchmark (AI-MO, 2024) with a maximum generation length of 32K and temperature 0.5 to prevent repetitive loops. All experiments are run on a single A100 80GB GPU, with further details in Appendix A.2.

### 4.1 MAIN RESULTS

#### 4.1.1 LONG SUMMARIZATION

Figure 4a shows that the Cross-model Retrieval cache substantially improves draft accuracy on long inputs, outperforming the static cache policy of StreamingLLM. This reduces the total number of draft-verify iterations, and combined with efficient attention mechanisms, leads to a significant reduction in inference time across all stages of speculative decoding (Figure 4b). As a result, SpecExtend achieves consistent speedup gains across all three datasets with both off-the-shelf LLMs and EAGLE draft models (Table 2).

For 8K and 16K-token inputs from PG-19, SpecExtend accelerates standard speculative decoding with LLM draft models by 2.37× and 2.22×, respectively, yielding overall speedups of 2.39× and 2.87× over naive autoregressive generation (Figure 5). For EAGLE-based frameworks, SpecExtend achieves 2.02× and 2.09× speedups over the standard EAGLE frameworks, yielding overall speedups of 2.67× and 3.09×. Importantly, SpecExtend preserves baseline performance on shorter inputs across all settings, demonstrating robustness to input length.

#### 4.1.2 LONG REASONING

Long reasoning has become a popular benchmark for testing LLMs on complex problem solving (DeepSeek-AI, 2025; Yang et al., 2024a). The task forces the model to handle *both* short and long sequences throughout generation. In this setting, while existing solutions like MagicDec fail to yield meaningful speedup on short inputs, a drop-in solution like SpecExtend is especially desirable, as it allows us to harness the strong short-input performance of SOTA frameworks.

As shown in Figure 6, SpecExtend improves draft accuracy by 3.15× over standard EAGLE-3, leading to a 3.86× speedup relative to the standard setup and a 3.73× speedup over naive autoregressive

| | Setting | SpecExtend | 1K $\tau$ | 1K Tok/s | 1K Speedup | 2K $\tau$ | 2K Tok/s | 2K Speedup | 4K $\tau$ | 4K Tok/s | 4K Speedup | 8K $\tau$ | 8K Tok/s | 8K Speedup | 16K $\tau$ | 16K Tok/s | 16K Speedup |
|---|---|---|---|---|---|---|---|---|---|---|---|---|---|---|---|---|---|
| GovReport V-7B | V-68M | No | 2.73 | 100.31 | 1.78× | 1.64 | 55.64 | 1.16× | 1.60 | 41.91 | 1.14× | 1.62 | 25.71 | 1.08× | 1.59 | 16.56 | 1.38× |
| | | Yes | **3.80** | **128.59** | **2.28×** | **3.52** | **109.58** | **2.29×** | **3.04** | **76.48** | **2.08×** | **3.06** | **55.16** | **2.33×** | **3.07** | **33.84** | **2.82×** |
| | EAGLE | No | **4.61** | 144.77 | 2.57× | 4.04 | 107.52 | 2.24× | 3.27 | 66.62 | 1.81× | 2.35 | 31.74 | 1.34× | 2.00 | 19.35 | 1.61× |
| | | Yes | 4.58 | **145.53** | **2.58×** | **4.08** | **113.47** | **2.37×** | **3.80** | **85.99** | **2.34×** | **3.82** | **62.90** | **2.66×** | **3.51** | **37.05** | **3.08×** |
| GovReport LC-7B | LC-68M | No | 2.73 | 100.31 | 1.78× | 1.64 | 55.64 | 1.16× | 1.60 | 41.91 | 1.15× | 1.62 | 25.71 | 1.12× | 1.59 | 16.56 | 1.51× |
| | | Yes | **3.01** | **109.26** | **1.94×** | **2.82** | **90.27** | **1.89×** | **2.66** | **68.84** | **1.89×** | **2.81** | **52.17** | **2.30×** | **2.68** | **31.11** | **2.84×** |
| | EAGLE | No | **4.10** | 131.06 | 2.33× | 3.47 | 97.53 | 2.04× | 2.75 | 60.17 | 1.65× | 2.52 | 32.90 | 1.44× | 2.18 | 19.84 | 1.81× |
| | | Yes | 4.04 | **133.14** | **2.37×** | **3.56** | **103.39** | **2.17×** | **3.43** | **80.50** | **2.21×** | **3.53** | **60.14** | **2.63×** | **3.25** | **35.13** | **3.21×** |
| PG-19 V-7B | V-68M | No | 2.16 | 76.50 | 1.37× | 1.52 | 51.00 | 1.09× | 1.55 | 39.16 | 1.15× | 1.55 | 21.80 | 1.01× | 1.54 | 14.73 | 1.29× |
| | | Yes | **2.75** | **96.74** | **1.74×** | **2.69** | **84.74** | **1.81×** | **2.61** | **63.94** | **1.88×** | **2.65** | **47.64** | **2.39×** | **2.70** | **32.88** | **2.87×** |
| | EAGLE | No | 3.29 | 107.31 | 1.92× | 3.18 | 88.92 | 1.89× | 2.88 | 54.71 | 1.60× | 2.18 | 26.43 | 1.32× | 1.92 | 16.98 | 1.48× |
| | | Yes | 3.29 | **107.53** | **1.93×** | **3.19** | **94.41** | **2.02×** | **3.04** | **69.92** | **2.06×** | **3.19** | **53.06** | **2.67×** | **3.05** | **35.43** | **3.09×** |
| PG-19 LC-7B | LC-68M | No | 2.16 | 76.50 | 1.36× | 1.52 | 51.00 | 1.07× | 1.55 | 39.16 | 1.09× | 1.55 | 21.80 | 1.00× | 1.54 | 14.73 | 1.18× |
| | | Yes | **2.22** | **80.25** | **1.43×** | **2.33** | **73.69** | **1.55×** | **2.42** | **62.27** | **1.74×** | **2.42** | **44.96** | **2.06×** | **2.45** | **30.67** | **2.46×** |
| | EAGLE | No | **3.19** | **111.10** | **1.97×** | 3.00 | 86.80 | 1.82× | 2.48 | 54.21 | 1.51× | 2.28 | 26.85 | 1.23× | 2.06 | 17.54 | 1.40× |
| | | Yes | 3.11 | 110.31 | 1.96× | **3.02** | **93.50** | **1.97×** | **2.97** | **71.84** | **2.01×** | **2.99** | **51.55** | **2.36×** | **2.82** | **33.07** | **2.66×** |
| BookSum V-7B | V-68M | No | 2.36 | 88.12 | 1.57× | 1.56 | 53.33 | 1.13× | 1.51 | 39.30 | 1.08× | 1.52 | 24.21 | 1.05× | 1.58 | 15.63 | 1.30× |
| | | Yes | **2.75** | **97.45** | **1.73×** | **2.66** | **81.37** | **1.73×** | **2.56** | **62.97** | **1.73×** | **2.70** | **50.21** | **2.18×** | **2.78** | **35.61** | **2.98×** |
| | EAGLE | No | **3.33** | 111.70 | **1.99×** | 2.95 | 82.44 | 1.75× | 2.87 | 58.01 | 1.59× | 2.14 | 29.30 | 1.27× | 1.94 | 18.76 | 1.57× |
| | | Yes | 3.31 | **111.82** | **1.99×** | **2.99** | **88.64** | **1.89×** | **3.08** | **70.90** | **1.95×** | **3.15** | **54.53** | **2.37×** | **3.11** | **38.03** | **3.18×** |
| BookSum LC-7B | LC-68M | No | 2.36 | 88.12 | 1.57× | 1.56 | 53.33 | 1.14× | 1.51 | 39.30 | 1.11× | 1.52 | 24.21 | 1.20× | 1.58 | 15.63 | 1.28× |
| | | Yes | **2.45** | **91.05** | **1.63×** | **2.55** | **83.60** | **1.80×** | **2.54** | **66.79** | **1.90×** | **2.61** | **49.47** | **2.45×** | **2.50** | **32.21** | **2.64×** |
| | EAGLE | No | **3.10** | **107.67** | **1.92×** | 2.94 | 86.35 | 1.85× | 2.37 | 53.42 | 1.51× | 2.22 | 30.14 | 1.49× | 2.06 | 18.39 | 1.50× |
| | | Yes | 3.07 | 106.86 | 1.91× | **2.97** | **90.48** | **1.94×** | **2.88** | **71.50** | **2.03×** | **2.92** | **52.35** | **2.59×** | **2.83** | **34.65** | **2.84×** |

Table 2: Average accepted length ($\tau$), decoding speed (tokens/s) and speedup of speculative decoding with and without SpecExtend. Speedup is measured relative to naive autoregressive generation.

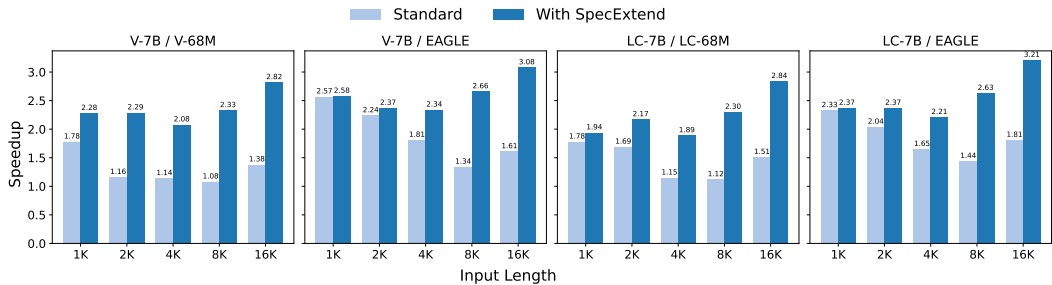

Figure 5: Speedup comparison of standard speculative decoding and SpecExtend across varying input lengths on GovReport.

decoding. We note that while EAGLE-3 achieves exceptional performance on short inputs, its draft accuracy drops sharply beyond 2K tokens, even falling below EAGLE-1 (Table 6). With SpecExtend, EAGLE-3 maintains high draft accuracy on long inputs while fully preserving its short-input strength, resulting in the substantial overall speedup.

## 4.2 COMPARISON WITH OTHER METHODS

We apply SpecExtend to standard speculative decoding and compare its performance on long inputs against other off-the-shelf acceleration methods, including FlashDecoding (Dao, 2024), TriForce (Sun et al., 2024), and MagicDec (Sadhukhan et al., 2024). For all frameworks, we use Vicuna-7B/68M as the target and draft models, respectively. For MagicDec, we implement StreamingLLM-based drafting with self-speculation. We exclude training-based methods (e.g., LongSpec) since SpecExtend is fully training-free, and its end-to-end performance depends heavily on the capacity and architecture of the draft model in the underlying framework.

| Method | GovReport 1K | 2K | 4K | 8K | 16K | PG-19 1K | 2K | 4K | 8K | 16K | BookSum 1K | 2K | 4K | 8K | 16K |
|---|---|---|---|---|---|---|---|---|---|---|---|---|---|---|---|
| FlashDecoding | 1.06× | 1.07× | 1.12× | 1.23× | 1.51× | 1.07× | 1.08× | 1.18× | 1.38× | 1.52× | 1.06× | 1.10× | 1.10× | 1.26× | 1.58× |
| TriForce | 1.25× | 1.26× | 1.22× | 1.18× | 1.02× | 1.12× | 1.19× | 1.16× | 1.15× | 1.13× | 1.18× | 1.20× | 1.18× | 1.18× | 1.11× |
| MagicDec | 1.07× | 1.08× | 1.05× | 1.13× | 1.24× | 1.03× | 1.07× | 1.06× | 1.10× | 1.19× | 1.03× | 1.04× | 1.06× | 1.18× | 1.23× |
| Standard | 1.78× | 1.16× | 1.14× | 1.08× | 1.38× | 1.37× | 1.09× | 1.15× | 1.09× | 1.29× | 1.57× | 1.14× | 1.08× | 1.05× | 1.30× |
| Standard + SpecExtend | **2.28×** | **2.29×** | **2.08×** | **2.29×** | **2.65×** | **1.74×** | **1.81×** | **1.88×** | **2.34×** | **2.70×** | **1.74×** | **1.74×** | **1.73×** | **2.14×** | **2.81×** |

Table 3: Speedup comparison of off-the-shelf methods for long sequence generation with Vicuna-7B. Standard refers to standard tree-based speculative decoding.

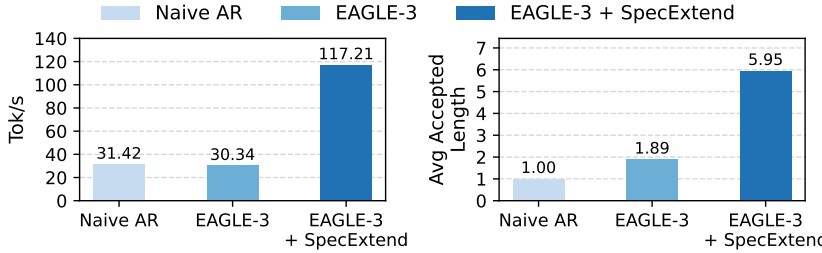

Figure 6: Decoding speed (left) and average accepted length (right) of the DeepSeek-R1-Distill-Llama-8B/EAGLE-3 setup on the long reasoning task with the AIME-24 benchmark.

As shown in Table 3, SpecExtend-enhanced speculative decoding outperforms all baselines across input lengths, achieving up to 2.81× speedup on 16K-token inputs from BookSum. In contrast, TriForce and MagicDec yield marginal speedups, as model weights remain the dominant memory bottleneck in moderately long regimes, yet both methods rely on drafting with the large base model.

## 4.3 ABLATION STUDIES

### 4.3.1 SPECEXTEND COMPONENTS

We evaluate the contribution of each component of SpecExtend with a standard Vicuan-7B/68M setup on GovReport. Speedups are reported relative to the standard setting. Table 4 shows that Cross-model Retrieval provides the largest gain, achieving a 1.46× speedup alone on 16K inputs, a 1.15× improvement over the static cache policy of StreamingLLM. FlashAttention applied to the prefill stages yields a 1.25× speedup. We note that Hybrid Tree Attention introduces minor overhead at shorter lengths but achieves up to 1.19× speedup beyond 8K tokens, thus we enable it only for inputs beyond 4K tokens.

| Setting | 1K | | | 2K | | | 4K | | | 8K | | | 16K | | |
|---|---|---|---|---|---|---|---|---|---|---|---|---|---|---|---|
| | $\tau$ | Tok/s | Speedup | $\tau$ | Tok/s | Speedup | $\tau$ | Tok/s | Speedup | $\tau$ | Tok/s | Speedup | $\tau$ | Tok/s | Speedup |
| Standard | 3.75 | 127.34 | - | 2.83 | 87.34 | - | 1.92 | 47.41 | - | 1.78 | 27.54 | - | 1.72 | 17.60 | - |
| Standard + FA | 3.71 | 131.02 | 1.03× | 2.84 | 91.79 | 1.05× | 1.97 | 52.74 | 1.11× | 1.81 | 34.33 | 1.25× | 1.75 | 22.07 | 1.25× |
| Standard + HTA | 3.61 | 122.73 | 0.96× | 2.74 | 85.57 | 0.98× | 1.92 | 47.62 | 1.01× | 1.76 | 31.08 | 1.14× | 1.74 | 20.95 | 1.19× |
| Standard + StreamingLLM | 3.75 | 128.62 | 1.01× | 2.81 | 85.60 | 0.98× | 2.53 | 59.11 | 1.25× | 2.59 | 35.89 | 1.30× | 2.60 | 22.39 | 1.27× |
| Standard + CMR | 3.86 | 130.35 | 1.02× | 3.57 | 104.12 | 1.19× | 2.90 | 64.85 | 1.36× | 2.78 | 37.11 | 1.47× | 2.93 | 25.82 | 1.46× |

Table 4: Ablation study of SpecExtend components. The standard setting refers to tree-based speculative decoding with Vicuna-7B/68M. FA denotes FlashAttention for prefill, HTA denotes Hybrid Tree Attention, and CMR denotes Cross-model Retrieval.

### 4.3.2 RETRIEVAL PARAMETERS

We ablate the parameters of Cross-model Retrieval using Vicuna-7B as the target model and Vicuna-68M/EAGLE as draft models on 8K-token GovReport inputs (Table 5). The optimal working KV cache size is around 1K for Vicuna-68M and 2K for EAGLE, which we adopt for the ablation. Under these settings, the best results are obtained with a chunk size of 32, top-k values of 32 and 64, and retrieval frequencies of 4 and 8 steps for Vicuna-68M/EAGLE, respectively.

| Working Cache Size | Vicuna-68M | EAGLE | Chunk Size | Vicuna-68M | EAGLE | Top-k | Vicuna-68M | EAGLE | Retrieval Frequency | Vicuna-68M | EAGLE |
|---|---|---|---|---|---|---|---|---|---|---|---|
| 64 | 32.52 | 39.10 | 1 | 31.05 | 48.05 | 2 | 30.72 | 38.22 | 1 | 33.05 | 47.78 |
| 128 | 32.91 | 39.95 | 2 | 32.27 | 49.49 | 4 | 32.65 | 40.36 | 2 | 33.54 | 46.78 |
| 256 | 33.65 | 41.53 | 4 | 32.97 | 49.55 | 8 | 32.76 | 41.49 | 4 | **33.59** | 48.17 |
| 512 | 33.53 | 42.77 | 8 | 33.39 | 49.18 | 16 | 33.19 | 43.90 | 8 | 33.11 | **48.52** |
| 1024 | **33.69** | 44.19 | 16 | 33.41 | 48.92 | 32 | **33.28** | 47.21 | 16 | 33.16 | 48.36 |
| 2048 | 32.36 | **45.33** | 32 | **33.52** | **49.68** | 64 | 32.50 | **48.09** | 32 | 33.28 | 48.11 |
| 4096 | 25.84 | 43.68 | 64 | 33.23 | 48.25 | 128 | 25.20 | 45.14 | 64 | 33.29 | 48.13 |
| 8192 | 24.32 | 33.10 | 128 | 33.20 | 47.48 | 256 | 23.95 | 32.48 | 128 | 33.21 | 48.20 |

Table 5: Ablation study of Cross-model Retrieval parameters. The table reports decoding speed (tokens/s) using Vicuna-7B as the target model on 8K-token GovReport inputs.

## 4.4 ADDITIONAL RESULTS

### 4.4.1 NEWER MODEL CONFIGURATION

We further demonstrate SpecExtend's compatibility by applying it to newer model configurations, using Llama-3.1-8B-Instruct as the base model with EAGLE and EAGLE-3 as draft models. EAGLE-3 introduces a modified draft architecture that enables larger-scale training. Although it achieves exceptional performance on short inputs, its accuracy degrades more sharply than EAGLE, with substantial performance drops even at 4K tokens (Table 6). With SpecExtend, EAGLE-3's draft accuracy improves by up to 2.55× on inputs of up to 16K tokens, yielding a 2.84× speedup over the standard setting and a 2.36× overall speedup. These results show that SpecExtend integrates seamlessly with newer speculative decoding frameworks.

| Draft Model | SpecExtend | 1K | | | 2K | | | 4K | | | 8K | | | 16K | | |
|---|---|---|---|---|---|---|---|---|---|---|---|---|---|---|---|---|
| | | $\tau$ | Tok/s | Speedup | $\tau$ | Tok/s | Speedup | $\tau$ | Tok/s | Speedup | $\tau$ | Tok/s | Speedup | $\tau$ | Tok/s | Speedup |
| EAGLE | No | 3.41 | 107.06 | 2.01× | 3.03 | 90.23 | 1.83× | 2.30 | 55.59 | 1.35× | 2.13 | 37.82 | 1.18× | 1.89 | 23.08 | 1.02× |
| | Yes | 3.42 | 108.01 | 2.04× | 3.10 | 91.68 | 1.87× | 3.02 | 69.02 | 1.68× | 2.92 | 51.33 | 1.60× | 2.78 | 41.66 | 1.85× |
| EAGLE-3 | No | 5.10 | 146.59 | 2.76× | 4.65 | 119.91 | 2.44× | 1.82 | 47.30 | 1.15× | 1.61 | 30.76 | 0.96× | 1.49 | 18.71 | 0.83× |
| | Yes | 5.03 | 145.65 | 2.75× | 4.68 | 120.35 | 2.45× | 3.99 | 89.93 | 2.18× | 3.96 | 66.52 | 2.08× | 3.80 | 53.18 | 2.36× |

Table 6: Evaluation of SpecExtend on LLaMA-3.1-8B-Instruct with EAGLE and EAGLE-3 on the GovReport dataset.

### 4.4.2 EXTREMELY LONG INPUTS

We also evaluate SpecExtend on sequences of up to 128K tokens using the Llama-3.1-8B-Instruct and EAGLE setup on PG-19. At this scale, the memory bottleneck shifts from model weights to the KV cache, making standard speculative decoding slower than naive autoregressive generation, since drafting becomes extremely slow even with a small draft model (Figure 2). By adopting the reduced Cross-model Retrieval cache, SpecExtend alleviates this bottleneck and also improves draft accuracy by 1.58×, yielding a 2.67× speedup over the standard setting (Table 7).

| SpecExtend | 32K | | | 64K | | | 128K | | |
|---|---|---|---|---|---|---|---|---|---|
| | $\tau$ | Tok/s | Speedup | $\tau$ | Tok/s | Speedup | $\tau$ | Tok/s | Speedup |
| No | 1.73 | 8.45 | 0.76× | 1.72 | 8.46 | - | 1.73 | 8.45 | - |
| Yes | 2.73 | 23.05 | 2.08× | 2.71 | 22.76 | - | 2.72 | 22.59 | - |

Table 7: Evaluation of SpecExtend on LLaMA-3.1-8B-Instruct with EAGLE for inputs up to 128K tokens on the PG-19 dataset. Naive autoregressive generation runs out of memory beyond 64K tokens, thus speedup values are omitted.

## 5 CONCLUSION

We presented SpecExtend, a drop-in enhancement that improves speculative decoding on long inputs. By combining efficient attention mechanisms with a novel KV cache eviction strategy, Cross-model Retrieval, SpecExtend accelerates all stages of speculative decoding while enhancing draft accuracy without retraining. Experiments show up to 2.84× speedup on long summarization and 3.86× on long reasoning, while preserving baseline performance on short inputs. SpecExtend is compatible with various speculative decoding setups and provides a practical, training-free solution to performance degradation on long inputs.

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

# A    APPENDIX

## A.1    LATENCY OVERHEAD OF CROSS-MODEL RETRIEVAL

|  | Target Forward | Target Forward w/ Retrieval | Draft Forward | Retrieval Cache Update |
|---|---|---|---|---|
| Latency (ms) | 53.76 | 54.11 | 0.84 | 0.34 |

Table 8: Latency overhead of a single retrieval cache update step on 16K token inputs.

## A.2    EXPERIMENT DETAILS

The EAGLE models[1] for vicuna-7b-v1.5-16k and longchat-7b-16k are trained on the ShareGPT dataset using default training settings with 4 A100 40GB GPUs. For each input length from 1K to 16K tokens, we sample 20 inputs, run each input twice, and report metrics averaged over all runs. We apply OPT-Tree's dynamic tree expansion strategy with the default settings of 50 total nodes, maximum depth 10, and threshold 0.7. We use the optimal working KV cache size and retrieval parameters described in Section 5.

---

[1]EAGLE models are publicly available under the Apache 2.0 license.

