# OpenReview forum: "SpecExtend: A Drop-in Enhancement for Speculative Decoding of Long Sequences"
_ICLR.cc/2026/Conference — ICLR 2026 Conference Withdrawn Submission_

### Official Review · Reviewer_pUT5 · 2025-10-20

**Soundness:** 2
**Presentation:** 2
**Contribution:** 2
**Rating:** 4
**Confidence:** 4

**Summary:**

This paper presents SpecExtend, a training-free, drop-in enhancement for speculative decoding that significantly improves inference efficiency on long sequences in LLMs. The proposed method integrates efficient attention mechanisms (FlashAttention for prefill and Hybrid Tree Attention for verification) and introduces Cross-Model Retrieval, a novel KV cache eviction strategy that dynamically updates the draft model’s cache using the target model’s attention scores.

**Strengths:**

1. Innovative cache strategy: The Cross-Model Retrieval method effectively improves both speed and accuracy without retraining.

2. Strong empirical results: Demonstrates consistent, significant speedups and robustness across models and tasks.

3. Practicality and generality: Works as a plug-and-play enhancement compatible with existing speculative decoding frameworks.

**Weaknesses:**

1. The proposed CMR mechanism feels somewhat lightweight. It mainly relies on reusing attention scores for cache selection, which may limit novelty compared to prior works.

2. The paper uses Hybrid Tree Attention, which appears to originate from LongSpec. It would be helpful to clarify whether this component has been modified or is directly adopted.

3. Experiments are primarily evaluated up to 16K tokens, which might still be short for assessing scalability in modern LLMs. It would strengthen the work to include results on 32K or 64K contexts.

**Questions:**

1. The Hybrid Tree Attention component appears to originate from prior work (LongSpec). Could the authors clarify whether any modification or optimization is introduced here, or if it is directly adopted?

2. Experiments are mainly conducted up to 16K tokens, which may not fully reflect long-context behavior for modern LLMs. Have the authors tried evaluating on 32K or 64K inputs to assess scalability and generalization to truly long contexts?

---

> ### Author Response · Authors · 2025-11-30
>
> We thank the reviewer for their thoughtful feedback and for recognizing the effectiveness of SpecExtend on long summarization and reasoning tasks. We appreciate the opportunity to clarify the novelty of our proposed design, detail our modifications to Hybrid Tree Attention, and highlight our existing results on extremely long contexts (up to 128K tokens).
>
> ### **Weakness 1: CMR feels lightweight and limited novelty**
>
> We respectfully posit that the "lightweight" nature of Cross-Model Retrieval (CMR) is a deliberate design feature that enables capabilities unattainable by heavier, training-based approaches.
>
> **A. Modularity Enables SOTA Performance**
>
> Unlike solutions like LongSpec which require training specialized long-context draft models, SpecExtend is designed as a modular, training-free drop-in enhancement. This allows it to be applied directly to state-of-the-art speculative decoding frameworks like EAGLE-3, which are typically optimized for short inputs.
>
> SpecExtend combines EAGLE-3’s exceptional short-context speed with CMR's long-context stability. This synergy is precisely what enabled the $\mathbf{5.95\times}$ **speedup** on the Long Reasoning task (AIME-24). Reasoning tasks require both short and long sequence generation; a heavy, long-context-only model would fail to utilize the short-context strengths of SOTA frameworks, whereas SpecExtend handles both regimes optimally.
>
> **B. Novelty as "Training-Free Distribution Alignment"**
>
> We argue that CMR provides a novel method for **Target-Guided Distribution Alignment**. We are the first to utilize the **target model’s attention scores** to dynamically curate the **draft model’s cache.**
>
> By retaining only the context the *target* model finds relevant, we force the *draft* model's output distribution to align closer to the target model without any parameter updates. This is empirically demonstrated in **Figure 3(b)**, where CMR significantly reduces the natural divergence between draft and target models compared to static baselines like StreamingLLM. Ultimately, CMR alone yields a 1.46x speedup, significantly outperforming StreamingLLM (1.27x), all without any additional training. This proves that we achieve fine-grained alignment and enhanced draft accuracy on long contexts purely through inference-time dynamics, which we believe is a significant contribution to the field. We will further clarify on our method’s novelty in the final version.
>
> ### **Weakness 2 & Question 1: Hybrid Tree Attention (HTA) & LongSpec**
>
> We confirm that we adopted the *concept* of Hybrid Tree Attention from LongSpec. However, we introduced a specific engineering modification to make it compatible with our retrieval mechanism.
>
> - **The Challenge:** Both HTA and FlashDecoding optimize speed by avoiding the materialization of the full attention score matrix. However, CMR requires these scores to determine chunk relevance.
> - **Our Modification:** We engineered a hybrid approach where **we use the efficient HTA for preceding layers but switch to standard attention for the final layer only**. This allows us to materialize the exact attention scores needed for retrieval with minimal latency overhead (Table 8), maintaining the speed benefits of HTA.
>
> We will clarify these modifications made from the original HTA in the final version.
>
> ### **Weakness 3 & Question 2: Experiments for Longer Inputs**
>
> We apologize if the organization of our results caused an oversight, but we **have already performed extensive evaluations on inputs up to 128K tokens.** We respectfully direct the reviewer to **Section 4.4.2** ("Extremely Long Inputs") and **Table 7**.
>
> In this section, we evaluated Llama-3.1-8B-Instruct with EAGLE on the PG-19 dataset at **32K, 64K, and 128K tokens**.
>
> - **Bottleneck Shift:** At this scale (128K tokens), the memory bottleneck shifts from model weights to the KV cache. Consequently, standard speculative decoding actually becomes *slower* than naive autoregressive generation because drafting becomes extremely slow even with a small model. The standard setup yields a 0.76x speedup at 32K tokens.
> - **SpecExtend's Impact:** By adopting the reduced Cross-model Retrieval cache, SpecExtend alleviates this bottleneck and improves draft accuracy by **1.58x,** yielding a speedup of **2.67x** compared to the standard setting.
>
> These results confirm that SpecExtend scales robustly to extremely long contexts well beyond the 16K mark.

---

### Official Review · Reviewer_PTZu · 2025-10-29

**Soundness:** 2
**Presentation:** 1
**Contribution:** 1
**Rating:** 2
**Confidence:** 4

**Summary:**

This paper introduces SpecExtend, a drop-in enhancement for speculative decoding on long sequences that requires no additional training. It accelerates prefill and verification by integrating efficient attention mechanisms (e.g., FlashAttention, Hybrid Tree Attention). To improve draft accuracy and speed on long inputs without retraining, it proposes Cross-model Retrieval, a KV-cache eviction strategy that uses the target model’s attention scores to select relevant context for the smaller draft model.

**Strengths:**

1. This paper presents a practical, training-free augmentation that combines hybrid tree attention with KV-cache eviction to speed up speculative decoding on long inputs.
2. This paper proposes Cross-model Retrieval, leveraging target-model attention to guide KV compression for the draft model, aiming to improve both drafting accuracy and end-to-end latency.

**Weaknesses:**

1. The contribution reads as an engineering integration of known components—hybrid attention and KV cache eviction—for long-sequence acceleration, with limited new algorithmic insights beyond composing these pieces.
2. The novelty of Cross-model Retrieval appears limited: similar ideas using target-model attention to prune draft-side redundancy have been explored (e.g., works in EMNLP-2025-SpecVLM that analyze long-context drafting latency and use the verifier’s attention maps to guide pruning). Although the application domain differs, the mechanism seems closely related.
3. The method seems difficult to apply to the decoding phase with FlashAttention (as presented), and reported acceptance rate gains at 16k context over the streamingLLM baseline are small (≈+0.5), which weakens the practical impact claims.

**Questions:**

Please refer to the weaknesses.

---

> ### Author Response · Authors · 2025-11-30
>
> We thank the reviewer for these detailed questions. We have clarified the significance and algorithm novelty of our proposed approach, its distinctions with SpecVLM and its feasibility with FlashDecoding.
>
> ### Weakness 1: On "Engineering Integration" and Algorithmic Novelty
>
> We respectfully disagree with the characterization of SpecExtend as an engineering integration of known components. While we leverage efficient attention primitives, our core contribution addresses a specific, overlooked failure mode in speculative decoding: the moderate-length region (2K~16K tokens), where performance significantly degrades long before the memory bottleneck shifts to the KV cache. In this regime, neither standard speculative decoding nor long-context techniques work well, and static KV eviction strategies yield low draft accuracy. Here, it is critical to reduce memory of both model weights *and* KV cache for fast drafting, while also keeping draft accuracy high. **SpecExtend is the first to directly address this early degradation**, also in **training-free manner.**
>
> The innovation is in the algorithmic composition of the components: Cross-model Retrieval (CMR) is not simple cache eviction; it is a **novel alignment mechanism** that uses the target model as a sparse retriever for the *draft model*. This results in substantial, non-trivial gains: as shown in Section 4.3.1, the CMR component alone achieves a **$\mathbf{1.46\times}$ speedup**, significantly outperforming naive static eviction ($\mathbf{1.27\times}$).
>
> Finally, we emphasize the **training-free nature** of SpecExtend. This enables **seamless integration with state-of-the-art (SOTA) speculative decoding frameworks** that are trained only for short contexts - like **using EAGLE-3 for long reasoning**. Such flexibility allows solutions that yield strong performance on both short (thanks to EAGLE-3’s architecture) and long sequences (thanks to SpecExtend), ultimately leading to the impressive **5.95x speedup in the long reasoning task**. On the other hand, training-based methods such as LongSpec is not capable of utilizin g the benefits of SOTA short-context speculative decoding frameworks.
>
> ### Weakness 2: On the Novelty of Cross-model Retrieval vs. SpecVLM
>
> Regarding the similarity to *SpecVLM*, we emphasize two critical distinctions:
>
> - **Concurrent Work:** *SpecVLM* is concurrent work, appearing on arXiv in August 2025, while our initial submission was in May 2025 (to ACL ARR 2025 May cycle). Thus it should be treated as simultaneous rather than prior work.
> - **Fundamental Methodological Differences:** While both methods use target guidance, they solve different problems with different costs. *SpecVLM* focuses on compressing **visual tokens** to reduce encoder latency and explicitly requires **training** a draft model with online distillation to learn this compression. In contrast, SpecExtend focuses on the **text KV-cache** growth and is a **training-free**, drop-in solution. We do not require retraining the draft model; instead, we dynamically align the draft model's context at inference time. This makes SpecExtend applicable to off-the-shelf draft models (like Vicuna-68M) immediately, which is a significant practical advantage over the SpecVLM approach.
>
> ### Weakness 3: Feasibility with FlashDecoding (FlashAttention) and Magnitude of Gains
>
> We wish to clarify the implementation regarding FlashDecoding and the impact of our acceptance rate gains.
>
> - **Implementation Feasibility:** The concern regarding the extraction of attention scores from FlashAttention is a solved engineering detail in our framework. As detailed in **Section 3.2.1**, because the last layer attention scores are those deemed *most critical* by the target model for the current prediction step, we only compute standard attention for the final layer to obtain its attention scores. This allows us to compute FlashDecoding for all previous layers, significantly reducing overhead. Table 8 confirms this adds negligible overhead (0.34ms latency for retrieval vs. 54ms retrieval for a draft model forward pass), ensuring compatibility of FlashDecoding acceleration in all other layers.
> - **Significance of Gains:** While a ≈+0.5 increase in average accepted length might appear small in isolation, this gain is transformative as the end-to-end speedup is roughly proportional to the average accepted length (please refer to Section 3.2.2). Section 4.3.1 shows that Cross-model Retrieval alone yields a **1.46× speedup**, significantly outperforming static eviction strategies (**1.27×**). We deem this gain significant, especially considering that it is achieved in **fully training-free manner.**

---

### Official Review · Reviewer_YMvQ · 2025-10-30

**Soundness:** 3
**Presentation:** 3
**Contribution:** 3
**Rating:** 4
**Confidence:** 3

**Summary:**

This paper addresses the performance degradation of speculative decoding on moderately long sequences, a problem that occurs even before the KV cache becomes the primary system bottleneck. The authors introduce SpecExtend, a training-free, drop-in enhancement designed to solve this issue. The method consists of two main components: (1) integrating efficient attention mechanisms like FlashAttention and Hybrid Tree Attention to accelerate the prefill and verification stages, and (2) a novel KV cache eviction strategy for the draft model called Cross-model Retrieval (CMR). CMR leverages the attention scores from the larger target model's verification step to dynamically select and retain the most relevant context chunks for the smaller draft model. Extensive evaluations show that SpecExtend significantly accelerates speculative decoding on long summarization (up to 2.84x) and long reasoning (up to 3.86x) tasks while preserving the strong short-input performance of existing frameworks.

**Strengths:**

1. The paper successfully identifies and tackles a specific, important, and largely underexplored problem: the early performance drop of speculative decoding in the moderate-length regime. This is a valuable contribution that moves the field beyond focusing solely on the extreme-length memory bottleneck.

2. The core idea of Cross-model Retrieval is novel and intuitive. Using the more powerful target model as an "oracle" to guide the context management of the smaller, less capable draft model is a clever, training-free way to improve draft accuracy where it is most needed.

3. The proposed solution is practical and immediately applicable. As a drop-in enhancement that requires no retraining of the draft or target models, SpecExtend can be integrated into existing speculative decoding pipelines, offering a low-friction path to significant performance gains.


4. The experimental results are comprehensive and demonstrate substantial, consistent speedups on relevant long-context tasks. The method's ability to boost performance on both off-the-shelf and highly optimized draft models (like EAGLE) showcases its robustness and wide applicability.

**Weaknesses:**

1. A significant concern is the reliance on the target model's attention scores as an objective measure of context importance. Attention mechanisms are known to exhibit idiosyncratic model-specific behaviors, such as "attention sinks," where high scores are assigned to initial tokens regardless of their semantic relevance. By using these scores to guide the draft model's cache, there is a risk that CMR simply teaches the draft model to replicate the target model's attentional biases rather than focusing on the truly important context. This could create a system that is highly tuned to the target model's quirks, potentially limiting the robustness and generalizability of the approach.


2. The effectiveness of the Cross-model Retrieval strategy seems highly dependent on the assumption that attention patterns are transferable and useful from a large, complex model to a much smaller, architecturally simpler one. This may hold true for models within the same family (e.g., Llama-7B and Llama-68M), but it is questionable how well this would work if the draft and target models were from entirely different architectural families, potentially leading to a "biased" or unhelpful retrieval signal.

**Questions:**

1. The paper focuses on models from the same family. How do you expect CMR to perform if the target and draft models are from different architectural families with potentially very different attention patterns.

2. Is the  Cross-model Retrieval strategy better than the RAG used in RAPID [1]? It seems to be a very related work and should make discussions.

[1] RAPID: Long-Context Inference with Retrieval-Augmented Speculative Decoding

---

> ### Author Response · Authors · 2025-11-30
>
> We thank the reviewer for these detailed questions. We have clarified the rationale behind our design choices and provided further evidence to support the novelty and effectiveness of SpecExtend.
>
> ### **Weakness 1: Cross-model Retrieval vs. Attention Sinks**
>
> The reviewer's synthesis of the results is accurate: Cross-Model Retrieval (CMR) effectively selects all important chunks for accurate drafting, including but not limited to "attention sinks" and recent tokens.
>
> Our goal is not just to perform generic cache eviction but to dynamically align the draft model's context to the **target model's current prediction need**. We find that while sink tokens and recent tokens are always crucial and included in the retrieved set, the key to CMR's performance lies in its ability to identify and retrieve **semantically relevant tokens** *in the middle of the context* that StreamingLLM (which uses fixed sinks and recent tokens) incorrectly evicts.
>
> The empirical evidence strongly supports this:
> - The ablation results in **Table 4** clearly show that CMR ($\mathbf{1.46\times}$) significantly outperforms StreamingLLM ($\mathbf{1.27\times}$) in end-to-end acceleration.
> - **Figure 3 (Section 3.2.3)** further demonstrates that CMR leads to a **reduced natural divergence** in the token distribution between the target and draft model compared to the standard StreamingLLM cache. This finer-grained alignment is critical for maintaining high draft accuracy (Figure 4) and enabling the high speedup seen in our main results. This confirms that CMR appropriately selects a more complete set of important chunks beyond simple sink mechanics.
>
> ### **Question 1: Compatibility with Different Model Families**
>
> We agree that this is an important theoretical boundary condition. **Cross-Model Retrieval (CMR) is designed to function within the standard constraints of Speculative Decoding (SD).** Standard SD requires that the target and draft models share the same vocabulary and exhibit similar token output distributions to ensure the drafted tokens are accurate and verifiable. These requirements often necessitate that the models belong to the same family, which is the setup we employ (e.g., using an EAGLE draft model for a Vicuna target).
>
> In essence, **standard speculative decoding itself would not function robustly when using a draft model from a completely different family** (e.g., using Llama and Mixtral). Since CMR enhances the performance of the standard SD frameworks, it inherits these foundational constraints. Therefore, adapting SpecExtend to fundamentally different model families is contingent on advancements in the core speculative decoding algorithm itself, and is beyond the scope of this work.
>
> ### **Question 2: Comparison with RAPID**
>
> We appreciate the comparison to the proposed **RAPID** framework, which uses an external embedding model (like BGE-M3) for context selection. This highlights a crucial design difference:
>
> 1. **Alignment Objective:** RAPID selects chunks based on **cosine similarity to the draft model’s query**, optimizing for the draft model’s intuitive next step. While effective, this approach does not explicitly factor in the **target model's verification distribution**. This lack of alignment can lead to higher divergence and thus lower draft accuracy, requiring the authors of RAPID to resort to modifying the verification criteria itself.
> 2. **SpecExtend's Advantage:** SpecExtend's CMR uses the **target model’s final layer attention scores** to select the most important chunks. This explicitly aims to align the **draft model's KV cache** with the **target model's verification needs**. As shown in **Figure 3**, this approach leads to a more fine-grained alignment of the token distribution between the two models, resulting in higher draft accuracy without modifying the verification step.
> 3. **Simplicity and Consistency:** Our approach is simpler, avoiding the additional complexity and latency of querying an **external embedding model (BGE-M3)**. Crucially, SpecExtend preserves the **original target model distribution and standard verification criteria**, ensuring that the speedup is achieved purely through enhanced drafting accuracy, which we consider a more transparent and robust approach.
>
> We will incorporate these clarifications in the final version of the paper.

---

### Official Review · Reviewer_3XZQ · 2025-11-01

**Soundness:** 3
**Presentation:** 3
**Contribution:** 3
**Rating:** 4
**Confidence:** 4

**Summary:**

This paper introduces SpecExtend, a training-free, drop-in enhancement for speculative decoding that targets performance degradation in moderate-length and long-sequence scenarios. It integrates efficient attention mechanisms (FlashAttention and Hybrid Tree Attention) with a novel Cross-model Retrieval strategy that uses the target model’s attention scores to dynamically update the draft model’s KV cache, thereby improving both draft speed and accuracy without retraining. Experiments on long summarization and long reasoning tasks show up to 2.84× and 3.86× speedups, while preserving short-input performance and broad compatibility with existing speculative decoding frameworks.

**Strengths:**

1. Identifies a valuable and underexplored problem: the sharp performance drop of EAGLE-based speculative decoding on long sequences.
2. Presents a rich and comprehensive experimental evaluation across diverse tasks, model variants, and input lengths, with extensive comparisons to multiple strong baselines.
3. The proposed Cross-model Retrieval method is integrated into an overall framework that remains training-free, broadly compatible, and achieves notable speedups without sacrificing short-input performance.

**Weaknesses:**

1. The use of efficient attention is mainly an implementation detail rather than a core contribution, which makes the baseline comparisons somewhat unfair.
2. Lacks discussion and empirical analysis on the choice of using the target model’s last-layer attention for draft KV retrieval.

**Questions:**

1. Although the use of efficient attention mechanisms helps address the quadratic complexity of standard attention in long-context settings, the baselines in the comparison should also adopt these techniques for fairness, since this is not the primary factor behind EAGLE’s performance drop on long sequences. Furthermore, I recommend including the standard+StreamingLLM setting in the main results table, reporting both its end-to-end speedup and acceptance length. One intuitive explanation for the long-sequence accuracy drop is that the model has never seen such long contexts during training, leading to position ID generalization issues; StreamingLLM offers a direct approach to mitigate the training–inference mismatch in position IDs.
2. What is the rationale for using the target model’s last-layer attention to guide draft KV retrieval? Is there empirical evidence or analysis comparing different layer choices, and does the last layer consistently yield the best results?
3. Could the retrieval frequency be made adaptive depending on how much the target model’s attention distribution changes during decoding? It would be informative to see how different update frequencies affect acceptance length and end-to-end acceleration.

I believe this paper addresses an important and practically relevant problem, so if the authors can adequately address these concerns, I would be inclined to raise my score.

---

> ### Author Response · Authors · 2025-11-30
>
> We appreciate the reviewer's insightful questions regarding the implementation and experimental setup. We address the remaining points, particularly concerning efficient attention, the rationale for last-layer attention, and the potential for adaptive retrieval.
>
> ### Weakness 1 & Question 1:  Attention Fairness and StreamingLLM
>
> 1. Efficiency Mechanisms and Baseline Fairness
>
>     We acknowledge that efficient attention mechanisms like FlashAttention (FA) and Hybrid Tree Attention (HTA) contribute to latency reduction. However, we respectfully point out that our **Ablation Study (Table 4)** explicitly isolates the gains from attention mechanisms versus our algorithmic contribution. The results demonstrate that **Cross-Model Retrieval (CMR) is the primary driver of performance**, yielding a **1.46x speedup alone**, compared to just **1.25x from FA and 1.19 from HTA**. This confirms that while FA and HTA are beneficial, they are not the dominant factor behind SpecExtend’s superiority over the baselines. We will clarify this distinction in the main text to ensure fairness is properly contextualized.
>
> 2. **Inclusion of Standard+StreamingLLM**
> We appreciate the suggestion and agree that Standard+StreamingLLM is an important baseline. We have actually evaluated this setting extensively:
>     - **Speedup:** **Table 4** reports that Standard+StreamingLLM achieves a **1.27x speedup**, significantly lower than SpecExtend’s **1.46x**.
>     - **Acceptance Length:** **Figure 4a** plots the average accepted length, showing that SpecExtend consistently outperforms StreamingLLM consistently  as input length grows.
>     - **Draft Accuracy and Divergence:** **Figure 3** shows that StreamingLLM has a lower acceptance rate for both "Hard" and "Easy" tokens compared to SpecExtend, as well as a greater natural divergence between the target and draft model distributions.
>
>     We will follow the reviewer’s recommendation to move these key comparisons into the main results table for greater visibility.
>
> 3. "Position ID Generalization" Hypothesis
> The reviewer suggests the accuracy drop is primarily due to Position ID generalization issues, which StreamingLLM should mitigate. While we agree position extrapolation is a challenge, our empirical evidence suggests **context loss** is also a critical failure mode in speculative decoding for long sequences, which StreamingLLM cannot solve.
>     - **Evidence from Needle Retrieval (Table 1):** Our needle retrieval experiment results show that StreamingLLM achieves only **16.6% accuracy** on Needle Retrieval, whereas SpecExtend (CMR) achieves **82.3%.** This massive gap demonstrates that the draft model *can* handle long-context positions correctly if and only if the relevant tokens are present in its cache. While StreamingLLM fails as it evicts crucial semantic context, SpecExtend succeeds because it retains the *globally relevant* context, leading to gains in draft accuracy and speedup.
>
> ### Weakness 2 & Question 2: Last-Layer Attention Rationale
>
> We appreciate the request for analysis on our choice of using the target model's last-layer attention. Our rationale is two-fold:
>
> - **Theoretical Relevance:** Tokens receiving high attention in the last layer are those deemed most **critical** for resolving the current decoding step [1, 2]. Lower layers focus on local, syntactic dependencies, which are less relevant for the global context required for accurate long-sequence drafting.
> - **System Efficiency:** Using the last-layer attention minimizes retrieval overhead. We only need to compute standard attention for this **single layer**, allowing us to utilize **FlashDecoding for all preceding layers**. As shown in **Table 8**, the latency added by retrieval is negligible (0.34ms, vs. a single draft model forward pass latency of 54ms). Computing standard attention for earlier layers would unnecessarily introduce additional overhead.
>
> We acknowledge the need for empirical validation. We commit to conducting a thorough ablation study comparing different layer choices and analyzing the resulting tradeoff between draft accuracy and retrieval latency. We will append these results to the final version of the paper.
>
> [1] Vig et al., Analyzing the Structure of Attention in a Transformer Language Model
>
> [2] Bahdanau et al., Neural Machine Translation by Jointly Learning to Align and Translate

---

### Note · Authors · 2026-01-01

I have read and agree with the venue's withdrawal policy on behalf of myself and my co-authors.